# Evaluation of the 22G Franseen needle and 22G Lancet needle for endoscopic ultrasonography-guided tissue acquisition sampling in solid pancreatic lesions: Propensity score weighting

Yuki Ishikawa-Kakiya, Hirotsugu Maruyama◉*, Kojiro Tanoue, Akira Higashimori, Masaki Ominami, Yuji Nadatani, Shusei Fukunaga, Koji Otani, Shuhei Hosomi◉, Fumio Tanaka, Yasuhiro Fujiwara

Department of Gastroenterology, Osaka Metropolitan University Graduate School of Medicine, Osaka, Japan

* hiromaruyama99@gmail.com

## Abstract

### Background and Aim

Advantages of endoscopic ultrasonography-guided tissue acquisition (EUS-TA) using a Lancet and Franseen needles have been evaluated and compared. However, little is known about the performance of each needle in diagnostic methods such as cytology and histology or the amount of blood contamination associated with each needle. This study aimed to compare the diagnostic yield and amount of blood contamination between two needles in patients with solid pancreatic lesions.

### Methods

We collected data of consecutive patients who underwent first time EUS-TA of solid pancreatic lesions at Osaka Metropolitan University between Jan 2006 and Jan 2021 from the electronic medical records. We compared the main outcomes (histological diagnostic accuracy) between the Lancet and Franseen needle groups. The amounts of core tissue and blood contamination were evaluated using a scoring system. This retrospective comparative study was conducted at a single center.

### Results

A total of 315 patients were divided into the Lancet (n = 200) and Franseen needle group (n = 115). The histological sensitivity and histological diagnostic accuracy of the Franseen needle was higher than that of Lancet needle (82.4% vs. 59.8%, respectively; odds ratio [OR], 2.29; 95% confidence interval [CI], 1.21–4.35; p = 0.011). Multivariate analysis using inverse probability of treat weighting method revealed that the diagnostic performance of the Franseen needle was the significantly higher than

**Data availability statement:** All relevant data are within the paper and its Supporting Information files.

**Funding:** The author(s) received no specific funding for this work.

**Competing interests:** The authors have declared that no competing interests exist.

the Lancet needle (OR, 2.74; 95% CI, 1.31–5.74; p = 0.007). The core tissue score of 4 was achieved in 53.3% of the Franseen group and 3.3% of the Lancet group (p < 0.001), while high blood contamination was observed in 53.3% and 40%, respectively (p = 0.089).

## Conclusion

The histological accuracy and the amount of tissue by the Franseen needle are higher than those of the Lancet needle. Franseen needle could achieve high histological diagnostic accuracy even with the same blood contamination rate as that in Lancet needle.

---

## Introduction

Pancreatic cancer has a poor prognosis and often progresses rapidly due to difficulties in its detection [1,2]. Endoscopic ultrasonography-guided tissue acquisition (EUS-TA) is an established diagnostic modality for cancer detection. Lancet-shaped needles have been widely used for EUS-fine needle aspiration (FNA); they are safe, and have high sensitivity, specificity, and diagnostic accuracy for pancreatic lesions [3–5]. However, some cases cannot be diagnosed due to insufficient tissue samples and blood contamination. The diagnosis in such cases is often delayed, causing patients to lose the opportunity for early treatment, which underscores the limitations of current diagnostic methods.

Newly shaped needles called "Franseen needles" have been developed for EUS-fine needle biopsy (FNB) [6]. According to the report by Itonaga et al and other some reports., the diagnostic accuracy of the Franseen needle was significantly higher than that of the standard needle, with an accuracy of 84.0% (95% confidence interval [CI]: 79.0–88.2) compared to 71.2% (95% CI: 65.2–76.6) (p < 0.001) [7,8]. Some studies have reported that the accuracy of the Franseen needle was significantly higher than that of Lancet needle [8–11] because of the higher amount of the core tissue. However, tissue obtained from EUS-FNAB is traditionally evaluated using cytology and histology, which has not been discussed about the details of diagnosis method such as cytological, histological or combined accuracy in the Franseen needle and Lancet needle. Moreover, it has been reported that the amount of core tissue collected with the Franseen needle is significantly larger than that collected with the Lancet needle, as demonstrated in multiple studies (P < 0.001). However, the amount of blood contamination between the Franseen needle and the Lancet needle has been directly compared in only a few studies [12]. In other words, the current literature lacks a comprehensive comparison of the histological accuracy of the Franseen and Lancet needles in pancreatic lesions, particularly with regard to blood contamination, which has not been adequately addressed in previous studies.

Here, we hypothesized that the histological accuracy of EUS-TA using a Franseen needle would be superior to that of a Lancet needle, and this factor would contribute to the total accuracy. We conducted a retrospective comparative study with the

aim to evaluate the histological accuracy of both needles in pancreatic lesions using the inverse probability of treatment weighting (IPTW) method [13]. In addition, we examined the amount of blood contamination, effect of which has not been sufficiently evaluated in previous studies.

## Methods

### Patient recruitment

A retrospective review of all patients who underwent EUS-TA for solid pancreatic lesions for the first time at Osaka Metropolitan University Hospital between Jan 2006 and Jan 2021 was conducted. Since 22G needles are generally used in Japan, the inclusion criteria were set as follows: patients with suspected solid pancreatic tumors who underwent EUS-TA using a 22G Lancet needle or a 22G Franseen needle for the first time. The exclusion criteria were cases with missing data, as mentioned in the data collection section. This retrospective comparative study was conducted at a single center.

### Ethical consideration

The study was conducted according to the Declaration of Helsinki. The study protocol was approved by the Ethics Committee of the Osaka City University Graduate School of Medicine (No. 2021–249, March 9, 2022). All the patients provided written informed consent for the use of their personal data. In addition, we provided the opportunity to opt out of the study on our website's homepage to all the patients. The data was accessed for research purposes from March 15, 2022 to September 30, 2024. During or after data collection, the authors had access to information that could identify individual participants for treatment purposes or otherwise.

### Data collection

Clinical information including age, sex, tumor size, lesion location, puncture route, needle type, number of punctures, endoscopists (trainee or expert), amount of core tissue and blood contamination, pathological diagnosis by EUS-TA, final diagnosis, and adverse events were collected from the patient electronic medical records (S 1 File). The collection of medical record information has been verified and is consistent with other doctors. We didn't had access to an anonymized patient dataset and conducted analyses based on specific patient information. No direct changes were made to individual data.

### Main outcome measure

The main outcome measure was the histological diagnostic accuracy of the 22G Lancet and 22G Franseen needles.

### Endoscopic procedure

All patients undergoing EUS-TA were administered an intravenous injection of midazolam (3–10 mg) and pentazocine (7.5–15 mg), with a dose depending on the age and tolerance. The procedures were performed using a curvilinear array echoendoscope (UCT240 and UCT260; Olympus Optical Corporation, Tokyo, Japan). Until the Franseen needle (Acquire® FNB needle, Boston Scientific Corporation, Natick, MA, USA) was commercially available, we used a Lancet needle (Expect®, Boston Scientific Corporation, Natick, MA, USA; SonoTip® ProControl, Medi-Globe GmbH, Rohrdorf, Bayern, Germany; EZ Shot3 Plus®, Olympus, Tokyo, Japan). After the Franseen needle was commercially available (starting December 2016), needle selection was based on an endoscopist's clinical judgement. First, in the absence of collateral vasculature, the tumor in pancreas was punctured using a needle. The stylet was removed from the needle and a 20 ml syringe was attached, which created a negative pressure at the site. Then, 20 to-and-fro movements were performed in the pancreatic tumor. We used different suction methods (general suction and wet-suction) to extract abundant good quality tissue and carry out the fanning technique if possible. After the passes, the needle was removed from the scope.

The stylet was reattached to the needle and injected a little saline was introduced into the needle assembly in a petri dish. During the procedure, cytologists performed rapid onsite evaluation (ROSE). When insufficient samples were obtained, the procedure was repeated. All patients were hospitalized a day after the procedure and physical characteristics were observed for occurrence of adverse events.

### Rapid onsite evaluation and histological assessment

We performed ROSE to improve pathological adequacy rates. A thin specimen was placed on a slide and air-dried by the endoscopist or cytotechnologist during the EUS-TA procedure. It was then stained with Diff-Quik and/or Papanicolaou to immediately evaluate sample adequacy and provide a preliminary diagnosis [14]. The cytological evaluation was based on the samples prepared by ROSE.

The remaining specimens were fixed in 10% buffered formalin solution for cell block preparation. Hematoxylin and eosin staining and immunostaining were performed, and a pathologist diagnosed the patients.

### Definitions

**Endoscopists.** The trainees and experts were divided as follows: Five years of experience with EUS and experience with >50 cases of EUS-TA was required to be qualified as an expert [8,12] and others were defined as trainees. At least one expert checked the examination in real time and performed the procedure instead of a trainee, if necessary.

**Malignant and benign lesions definitions.** We defined pancreatitis, intraductal papillary mucinous neoplasm (IPMN), intraductal papillary mucinous adenoma, serous cystic neoplasm, neuroendocrine tumor (G1, G2), autoimmune pancreatitis, pseudopancreatic cysts, accessory spleen, and lymphoepithelial cysts as benign. We defined high-grade dysplasia and invasive carcinoma IPMN, neuroendocrine tumor (G3), malignant lymphoma, metastatic pancreatic cancer, carcinoma of unknown primary origin, and pancreatic cancer as malignant [15].

**True and false diagnosis.** An adequate sample was defined as the percentage of lesions sampled in which the obtained material is representative of the target site and sufficient for diagnosis. Then, the rest were defined as inadequate samples [15,16].

The final diagnosis was based on surgical histological results in patients who underwent surgery. In case where patients did not undergo surgery, positive EUS-TA results, disease progression, along with a follow-up period of 12 months, were considered the final diagnosis [17,18]. True diagnosis was defined as the coinciding of the final diagnosis and EUS-TA results. Different results were defined as a false diagnosis. Diagnostic accuracy was defined as the percentage of lesions that obtain true diagnosis.

**Adverse events.** Adverse events were evaluated according to the American Society for Gastrointestinal Endoscopy workshop report [19].

**Amount of core tissue and blood contamination.** It was not possible to collect slides for all cases, therefore we randomly extracted 30 cases from the available slides. We assigned serial numbers to the remaining slides from both the Lancet needle group and the Franseen needle group. Then, we generated a random number table in Excel and selected 30 cases starting from the smallest numbers. The amount of core tissue was scored as: 1 (no material), 2 (a tissue fragment), 3 (a small histological core tissue <×10 objective), and 4 (a large histological core tissue >×10 objective); and blood contamination as: 1 (few), 2 (moderate), and 3 (high) (Fig 1) [20].

### Statistical analyses

If there was missing data, those cases were excluded. Continuous variables are presented as mean ± standard deviation or median ± interquartile range, while categorical variables as numbers. The data were evaluated using unpaired t-tests (continuous variables) or the Fisher's exact test (categorical variables). The Kolmogorov-Smirnov test for normality

|  | Amount of core tissue | Amount of blood |
|---|---|---|
| Score 1 | None | None - a little |
| Score 2 | Fragmented tissue | Moderate |
| Score 3 | Small pieces of tissue under 10x field of view | Large |
| Score 4 | Large pieces of tissue with a 10x field of view or more | - |

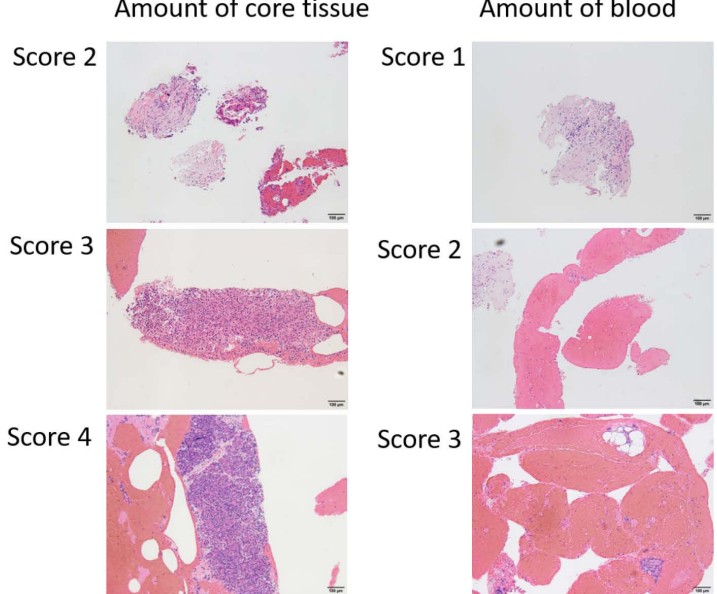

**Fig 1. Definition of core tissue and blood contamination scores.** ‡‡: EUS-TA, endoscopic ultrasound guided fine-needle aspiration biopsy.

indicated that age, size, and the number of punctures did not follow a normal distribution; therefore, the Mann-Whitney U test was used for analysis. If the contingency tables are larger than 2x2, the Fisher-Freeman-Halton test was used. The model included factors such as age, sex, tumor size, lesion location, puncture route, type of needle, number of punctures, endoscopist, rate of adequate sample, pathological diagnosis by EUS-TA, final diagnosis, and adverse events. To assess the association of prognostic factors for histological diagnostic accuracy were identified by univariate logistic regression analyses, odds ratio (OR) with a 95% confidence interval (CI) was calculated. Number of punctures and puncture route, a factor associated with diagnostic accuracy, were analyzed in the multivariate analysis [8,11,17,21]. Furthermore, the amount of tissue and blood contamination and diagnostic accuracy for benign and malignant diseases was analyzed using Fisher's exact test.

For the semi-quantitative analysis of the amount of core tissue and blood contamination, we corrected 30 cases from each group using a computer-generated random number table. Kappa coefficients were calculated to assess the inter- and intra-observer agreement between the amount of core tissue and blood contamination. A κ-value <0.50 was regarded as poor agreement, between 0.5 and 0.75 as moderate, between 0.75 and 0.90 as good, and > 0.90 as excellent [22,23].

Further, IPTW based on propensity scores were used to reduce selection bias by creating a "pseudo-population" in this study. The IPTW was calculated as the inverse of the conditional probability of receiving the specific exposure, defined here as the type of needle (Franceen or Lancet) used during the EUS-TA procedure, that the patients received [24,25].

Variables (age, sex, tumor size, lesion location, puncture route, and endoscopist) that might influence the diagnostic accuracy were used to generate a propensity score using logistic regression analysis. The reliability of the model was evaluated using the Hosmer–Lemeshow goodness-of-fit statistical analysis.

Statistical analyses were performed using the SPSS™ software (version 26.0; SPSS Inc., Japan) and the R™ statistical package V2.9-1 (http://www.r-project.org). All statistical tests were two-sided, and *p*-values <0.05 were considered significant.

## Results

### Baseline characteristics of patients

We collected data from 365 patients with solid pancreatic lesions who underwent EUS-TA using a 22G needle and excluded 50 patients: 1 from the Franseen needle group (could not be followed up) and 49 from the Lancet needle group (loss of data regarding the location, puncture route, number of punctures, endoscopists, and follow up).

A total of 315 patients were enrolled in this study (Fig 2). The clinical characteristics of the patients are shown in Table 1. The patients were classified into two groups: the Lancet needle group (n = 200) and Franseen needle group (n = 115). The median age was 69 years in the Franseen needle group and 71 years in the Lancet needle group, with the majority of participants being men (57.5% vs. 53.9%), showing no significant difference. The tumor size was significantly larger in the Franseen needle group than in Lancet needle group (p < 0.001). The number of expert endoscopists was significantly higher in the Lancet needle group than in Franseen needle group (p < 0.001). The median number of punctures was two in both groups. There was no significant difference in the occurrence rate of benign and malignant tumors between the two groups. The adequate specimen acquisition rate was 91% in the Lancet needle group and 96.5% in the Franseen needle group, with a significant difference observed between the two groups (P = 0.069).

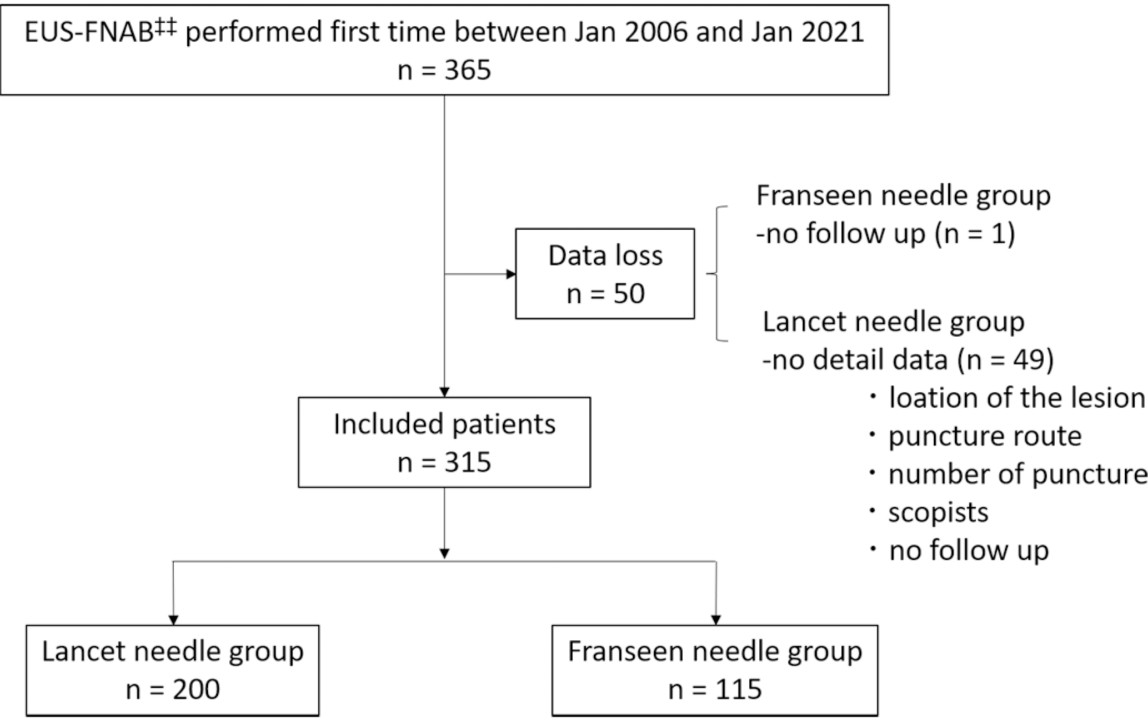

**Fig 2. Scheme of the study design.**

**Table 1. Clinical characteristics of the Lancet and Franseen needle groups.**

| | Lancet needle (n = 200) | Franseen needle (n = 115) | p value |
|---|---|---|---|
| **Median age (IQR†), years** | 69 [60–74] | 71 [66–76] | 0.009 |
| **Sex, male, n (%)** | 115 (57.5) | 62 (53.9) | 0.557 |
| **Median tumor size, mm (IQR†)** | 22 [16.1–27.9] | 27 [20.9–32.8] | <0.001 |
| **Location of the lesion, n (%)** | | | 0.008 |
| **Head** | 119 (59.5) | 52 (45.2) | |
| **Body** | 57 (28.5) | 34 (29.6) | |
| **Tail** | 22 (11.0) | 28 (24.3) | |
| **All** | 2 (1.0) | 1 (0.9) | |
| **Puncture route, n (%)** | | | 0.001 |
| **Stomach** | 85 (42.5) | 68 (59.1) | |
| **Duodenal bulb** | 57 (28.5) | 31 (27.0) | |
| **Duodenal second portion** | 58 (29) | 15 (13.0) | |
| **Jejumum** | 0 | 1 (1.0) | |
| **Scopist, expert (%)** | 65 (32.5) | 11 (9.6) | <0.001 |
| **Median number of passes (IQR†)** | 2 [2–3] | 2 [1–2] | 0.003 |
| **Adverse events, n (%)** | 3 (1.5) | 3 (2.6) | 0.672 |
| **Infection** | 3 (1.5) | 0 | |
| **Pancreatitis** | 0 | 3 (2.6) | |
| **Surgery** | 86 (43.0) | 34 (29.5) | 0.021 |
| **Final diagnosis, n (%)** | | | |
| **Malignant** | 139 (69.5) | 91 (79.1) | 0.067 |
| **Pancreatic carcinoma** | 126 | 82 | |
| **Neuroendocrine carcinoma** | 1 | 1 | |
| **Metastatic cancer** | 5 | 4 | |
| **Intraductal papillary mucinous with high-grade dysplasia** | 1 | 0 | |
| **Intraductal papillary mucinous with invasive carcinoma** | 1 | 2 | |
| **Malignant lymphoma** | 4 | 1 | |
| **Carcinoma of unknown primary** | 1 | 1 | |
| **Benign** | 61 (30.5) | 24 (20.9) | |
| **Autoimmune pancreatitis** | 12 | 11 | |
| **Neuroendocrine neoplasm** | 15 | 5 | |
| **Tumor-forming pancreatitis** | 2 | 0 | |
| **Serous cystic neoplasm** | 5 | 1 | |
| **Non-specific inflammation** | 14 | 2 | |
| **Chronic pancreatitis** | 6 | 2 | |
| **Intraductal papillary mucinous neoplasm** | 2 | 2 | |
| **Pancreatic pseudocyst** | 2 | 0 | |
| **Lymphoepitelial cyst** | 0 | 1 | |
| **Intrapancreatic accessory spleen** | 3 | 0 | |
| **Adequate sample, %** | 182 (91.0) | 111 (96.5) | 0.069 |

†: IQR, interquartile range

## Main outcome measurements

**Histology.** We excluded cases with histologically inadequate specimens (22 cases), cytologically inadequate specimens (8 cases), and cases where no specimens were submitted (13 cases) from the study population and performed the analysis for the main outcomes. The percentage of sensitivity, specificity, positive predictive value, negative predictive value, and adequate sample size was higher for the Franseen needle than for Lancet needle. The histological diagnostic accuracy of the Franseen needle was higher than that of the Lancet needle (84.7% and 70.7%, respectively; p = 0.026) (Table 2).

**Cytology/ Combination of cytology and histology.** No significant difference between the needles in the accuracy of the cytological evaluation was observed (85.7% and 82.8%, respectively; p = 0.696). The combined (cytology and histology) diagnostic accuracy did not show a significant difference between the Franseen and Lancet needles (92.9% and 86.2%, respectively; p = 0.165) (S1 Table).

**Prognostic factor analyses for histological diagnostic accuracy.** Table 3 shows the results of the univariate analysis of histological diagnostic accuracy. The histological diagnostic accuracy of the Franseen needle was significantly higher than that of Lancet needle (OR, 2.29; 95% CI, 1.21–4.35; p = 0.011). Additionally, the univariate and multivariate analysis revealed that the use of the number of punctures was a prognostic factor for increased histological diagnostic accuracy (OR, 1.47; 95% CI, 1.14–1.88; p = 0.003) (OR, 1.35; 95% CI, 1.04–1.75; p = 0.022). Age, gender, tumor size, expert, and puncture route did not influence histological diagnostic accuracy.

**Table 2. Histological diagnostic abilities of a Lancet and Franseen needle.**

|  | Lancet needle (n = 174) | Franseen needle (n = 98) | p value |
|---|---|---|---|
| Sensitivity, % (95% CI) | 59.8 (0.51-0.68) | 82.4 (0.73-0.90) | <0.001 |
| Specificity, % (95% CI) | 100 (0.89-1.00) | 100 (0.66-1.00) | 1 |
| PPV‡, % (95% CI) | 100 (0.93-1.00) | 100 (0.92-1.00) | 1 |
| NPV§, % (95% CI) | 48.0 (0.38-0.58) | 46.4 (0.28-0.66) | 0.887 |
| Diagnostic accuracy, % (95% CI) | 70.7 (0.63-0.77) | 84.7 (0.76-0.91) | 0.026 |

§: NPV, Negative predictive value; ‡: PPV, Positive predictive value

**Table 3. Prognostic factor for histological diagnostic accuracy.**

|  | Univariate | | Multivariate | |
|---|---|---|---|---|
|  | Crude OR¶¶ (95% C\|\|\|\|) | p value | Crude OR¶¶ (95% CI\|\|\|\|) | p value |
| **Age** | 1.00 (0.98-1.03) | 0.656 |  |  |
| **Sex, male** | 1.03 (0.59-1.81) | 0.911 |  |  |
| **Tumor size** | 1.00 (0.97-1.02) | 0.767 |  |  |
| **Expert** | 1.21 (0.64-2.29) | 0.566 |  |  |
| **Puncture route, duodenum** | 1.07 (0.53-0.77) | 0.122 | 1.17 (0.83-1.65) | 0.379 |
| **Number of punctures** | 1.47 (1.14-1.88) | 0.003 | 1.35 (1.04-1.75) | 0.022 |
| **Needle, Franseen** | 2.29 (1.21-4.35) | 0.011 | 1.85 (0.95-3.61) | 0.069 |
| **Cut-off value, AUC (95% CI)** |  |  |  |  |
| Number of punctures | 3, 61.9 (0.54-0.69) |  |  |  |
| Tumor size, (mm) | 29, 51.0 (0.44-0.58) |  |  |  |

¶¶: OR, odd ratio, \|\|\|\|: CI, confidence interval

**Table 4. Prognostic factors of histological diagnostic accuracy analyzed by IPTW††.**

| | Odds ratio (95% CI||||) | p value |
|---|---|---|
| **Before IPTW††** | | |
| Needle, Franseen | 2.29 (1.21–4.35) | 0.011 |
| Adjusted for number of punctures, puncture route | 1.85 (0.95–3.61) | 0.069 |
| **After IPTW††** | | |
| Needle, Franseen | 3.09 (1.51–6.29) | 0.002 |
| Adjusted for number of punctures, puncture route | 2.74 (1.31–5.74) | 0.007 |

††: IPTW, inverse probability of treatment weighting, ||||: CI, confidence interval

**Table 5. Semi-quantitative analyses of histological specimen according to needle type.**

| | Lancet needle (n = 30) | Franseen needle (n = 30) | p value |
|---|---|---|---|
| Core tissue score of 1–3, % | 96.7 | 46.7 | <0.001 |
| Core tissue score of 4, % | 3.3 | 53.3 | |
| Blood contamination score of 1–2, % | 60 | 46.7 | 0.089 |
| Blood contamination score of 3, % | 40 | 53.3 | |

Additionally, IPTW method was used to reduce the selection bias; the analysis revealed that the Franseen needle was a prognostic factor for increased diagnostic accuracy after adjusting for the number of punctures and puncture route (an influencing factor for diagnostic accuracy), (OR, 2.74; 95% CI, 1.31–5.74; p = 0.007) (Table 4). Evaluation of the propensity-weighted model was well calibrated (Hosmer–Lemeshow test, p = 0.36).

**The amount of core tissue and blood contamination.** The percentage of extracted samples with a core tissue score 4 was higher with the Franseen needle than with Lancet needle (p < 0.001). Blood contamination score of 3 was observed in a higher percentage of samples extracted with the Franseen needle than in that with Lancet needle; however, there was no significant difference (p = 0.089) (Table 5).

We evaluated the inter- and intra-observer agreement between the two endoscopists (Y. K. and H. M.) for the amount of tissue and blood contamination. We confirmed poor inter-and agreement (κ-value = 0.22, 95% CI: 0.02–0.41) and moderate intra-observer agreement (κ-value = 0.54, 95% CI: 0.39–0.69) for the score.

**Histological diagnostic accuracy for malignancy.** There was no significant difference between the Lancet and Franseen needles in terms of histological diagnostic accuracy limited to benign tumors (100%, and 100%, respectively) (p = 1.000). In contrast, the Franseen needle's histological diagnostic accuracy limited to malignant tumors (82.4%, and 59.8%, respectively) (p < 0.001) was superior to that of the Lancet needle. Cytological and histological diagnostic accuracy was also superior than Lancet needle (91.8%, and 81.1%, respectively) (p = 0.046) (S2 Table).

**Adverse events among the study subjects.** The rate of adverse events was 1.5% (infection, 3) in the Lancet needle group and 2.6% (pancreatitis, 3) in the Franseen needle group (p = 0.672) (Table 1). All patients were treated with conservative therapy. Needle tract seeding was not observed.

## Discussion

The histological diagnostic accuracy of the Franseen needle was significantly higher than that of Lancet needle (OR, 2.74; 95% CI, 1.31–5.74; p = 0.007) using IPTW, and Franseen needle was found to be more adequate and collect more tissue than the Lancet needle (96.5% vs 91.0%). It was suggested that the improved accuracy of the Franseen needle was owing to its ability to obtain much accurate sample. This result could be important not only for pancreatic ductal

adenocarcinoma but also for atypical pancreatic tumors, such as acinar cell carcinoma, anaplastic carcinoma, and neuro-endocrine tumors, and can have a positive impact on needle selection.

A three-pointed crown-shaped cutting heels needle (Franseen needle) for EUS-FNB developed to extract larger pieces of tissue has been demonstrated to have a high diagnostic accuracy compared with that of a Lancet needle. The Franseen needle could extract a larger diagnostic area [26] and more adequate tissue than the standard needle [15], contributing to higher diagnostic accuracy. Almost all previous studies concluded that Franseen needle obtained higher amount of tissue, which led to its high diagnostic accuracy. However, the diagnostic methods such as cytology or histology were not discussed when evaluating the performance of the Franseen needle. The FNB needle was made as a biopsy needle; however, we traditionary evaluated the tumor using "cytology" and "histology". It is unclear which diagnostic technique influenced the final diagnosis. Therefore, we analyzed the accuracy of each type of diagnosis (cytology, histology, and cytology and histology combined) to establish the diagnostic efficacy of the Franseen needle and concluded that Franseen needle's histological diagnosis was superior to that of the Lancet needle (Tables 3 and 4).

Previous reports have indicated that the number of punctures and puncture route are factors that influence diagnostic accuracy [8,11,20,21]. In the present study, we adjusted for these factors using multivariate analysis, and we found that the diagnostic accuracy of the Franseen needle was still higher compared to the lancet needle. Regarding tumor size and diagnostic accuracy, some studies have shown that diagnostic accuracy is higher in larger tumors [20], however, others have found no significant difference. According to the AGA White Paper, tumor sizes exceeding 20 mm with four punctures can achieve high diagnostic accuracy [17]. However, in our study, tumor size did not affect diagnostic accuracy. This may be attributed to the number of punctures or puncture techniques (e.g., fanning or door-knocking methods), which were not evaluated in this retrospective study. Subsequently, we evaluated the impact of expert experience on diagnostic accuracy. Previous reports suggest that the skill level of the endoscopist does not significantly affect diagnostic accuracy [27]. Similarly, our findings were consistent with this. One possible reason for this is that all procedures at our institution are performed under the supervision of an expert.

We hypothesized that two factors influenced the Franseen needle's excellent histological diagnosis: First, the Franseen needle tended to extract an adequate sample in 96.5% of histological evaluations, whereas Lancet needle extracted the same type of material in 91.0% (p = 0.069) of cases, and the tissue score was higher in the Franseen needle group than in Lancet needle group (p < 0.001) (Tables 1 and 5). This finding is similar to those reported in previous studies [12,15]. Inadequate tissue was extracted by the Lancet needle in cases of pancreatic ductal adenocarcinoma, autoimmune pancreatitis, malignant lymphoma, metastatic pancreatic tumor, and serous cystic neoplasms. Concerns arose about samples collected using the Lancet needles suggesting that they would consist of insufficient tissue and excessive amount of blood, which may be inadequate for detecting low epithelial and atypical pancreatic tumor. Second, the amount of tissue is important for diagnosis. Some studies have reported that EUS-FNB using a Franseen needle extracted a large tissue sample containing desmoplastic fibrosis [6,8,10,26]. A large amount of tissue provides more information for pathologists. A small tissue sample of fragmented atypical epithelial clusters may cause difficulties to diagnose pancreatic acinar or gastric foveolar epithelial cells due to inflammation. Furthermore, the desmoplastic reaction helps in diagnosis because pancreatic ductal adenocarcinoma invasively proliferates accompanying desmoplastic reactions. This reaction is useful in diagnosis, even for inexperienced pathologists. In this study, similar to previous studies [8], semiquantitative analysis showed that the amount of core tissue extracted using the Franseen needle was higher than that using Lancet needle. In the sub-group analysis, the Franseen needle's histological diagnostic accuracy limited to malignant lesions was higher than that of the Lancet needle and there was no significant difference when the accuracy was limited to benign lesions. This suggests that large core tissues contribute to the diagnostic accuracy of histological information such as desmoplastic reactions.

Franseen needles could obtain not only core tissue but also contribute to the blood contamination; however, blood contamination score showed no significant difference between the two groups (p = 0.089) (Table 5). This result is very

important for selecting Franseen needles for various tumors. The blood contamination should be minimized because it affects diagnostic interpretation. The previous meta-analysis did not reveal a statistically significant difference in blood contamination using the wet-suction vs. dry-suction [16,18]. Therefore, the use of the suction or non-suction method appears to strongly influence the amount of blood contamination [18,22,28]. Ishigaki et al. reported that, compared with the suction group, a blood contamination score of ≥ 2 was seen in a lower percentage of the no-suction group. Therefore, avoiding suction could reduce blood contamination [12]. This study compared the amount of blood contamination in Lancet and Franseen needles using suction method alone. This suggests that the needle shape alone does not affect the amount of blood contamination.

Contrary to the histological results, the Franseen needle did not have better cytological diagnostic accuracy. Pancreatic duct cancers are moderately to well differentiated, and cell atypia is inconspicuous. Therefore, even if a large sample is obtained, diagnosis is difficult in some cases, and they require extensive information such as desmoplastic reactions and histological immunostaining for diagnosis. Furthermore, the technical problems of cytotechnologists and resistance to diagnosing malignant lesions may also have an effect. To increase cytological diagnostic accuracy, it might be necessary to improve the quality of the cytotechnologist and well-designed cell blocks or liquid-based cytology, rather than the quantity of the sample.

Adverse events, such as hematoma, bleeding, and pancreatitis, after EUS-TA with a Franseen needle have been reported; however, the incidence rate is low [7,11,28,29]. In this study, only a few adverse events occurred in each group, and these patients were treated conservatively. Needle tract seeding of the Lancet and Franseen needles did not occur. However, Yane et al. reported that the risk of Franseen needle tract seeding after EUS-TA was 3.4% [30]. Therefore, careful follow-up is required after EUS-TA for the pancreatic body and tail lesions.

This study has few strengths. First, this study sample included many patients who underwent EUS-TA using a Franseen needle. Second, we conducted quasi-randomized studies using the IPTW method, which improved the study's quality. Nonrandomized controlled trials typically use a multiple regression model; however, in recent years, propensity scores have become more popular. Although the propensity score method cannot replace randomized trials, it is a useful alternative that has advantages over traditional methods [31]. Third, we investigated the relationship between the amount of blood contamination and the needle type.

This study had few limitations. First, we could not evaluate the relationship between factors such as the utilization of suction method (general suction or wet-suction), however, these factors may potentially influence diagnostic accuracy [32,33]. Previous reports have indicated that wet suction resulted in blood contamination similar to that of the general suction method, but yielded better results in terms of diagnostic rate and tissue quality [32,34]. Similarly, the puncture technique may affect diagnostic accuracy; however, due to insufficient data on this factor, it could not be fully evaluated in this study. Second, the study utilized three types of needles—EZ Shot 3 Plus, Expect, and ProCore—which were selected based on their availability and widespread use during the study period (2006–2021). However, differences in manufacturing processes, quality, and design among these needles may have influenced the results, including diagnostic yield and sample adequacy. Variations in technological advancements over time, such as changes in material composition or needle tip design, could also have introduced performance differences that were not fully controlled in this study. Third, since many older cases were included, there may have been slight differences in technique and diagnostic ability. Forth, we evaluated only 30 randomized samples from each group. We could not correct all pathological samples for semi-quantitative analysis of core tissue and blood contamination scores because they were too old to correct the full sample retrospectively. Fifth, this was a retrospective study that used the IPTW method. Propensity score analysis is a statistical method for adjusting for selection bias in observational studies and approximates randomized trial approaches. However, adjusting for potential confounders using propensity score matching analysis is difficult. Sixth, the study was conducted at a single facility in Japan, and there is a possibility that it may not be representative of the entire population.

## Conclusions

Our retrospective analysis indicated that the newly developed Franseen needle for EUS-TA was associated with a higher histological diagnostic accuracy and lower amount of blood contamination than Lancet needle did for pancreatic lesions. Therefore, utilization of a Franseen needle for EUS-TA might be advantageous in a clinical setting to achieve accurate diagnosis.

## Supporting information

**S1 File. Patient data used for analysis.** Includes anonymized demographic and clinical information.
(XLSX)

**S1 Table. Diagnostic abilities of a Lancet and Franseen needle.**
(DOCX)

**S2 Table. The diagnostic performance for malignant/benign lesions according to the needle type.**
(DOCX)

## Acknowledgments

We would like to thank Editage (www.editage.jp) for the English language editing.

## Author contributions

**Conceptualization:** Yuki Ishikawa-Kakiya, Hirotsugu Maruyama.

**Data curation:** Yuki Ishikawa-Kakiya, Hirotsugu Maruyama.

**Formal analysis:** Yuki Ishikawa-Kakiya, Hirotsugu Maruyama.

**Investigation:** Yuki Ishikawa-Kakiya, Hirotsugu Maruyama.

**Methodology:** Yuki Ishikawa-Kakiya, Hirotsugu Maruyama.

**Project administration:** Yuki Ishikawa-Kakiya, Hirotsugu Maruyama.

**Resources:** Yuki Ishikawa-Kakiya.

**Supervision:** Yasuhiro Fujiwara.

**Validation:** Yuki Ishikawa-Kakiya, Hirotsugu Maruyama.

**Visualization:** Yuki Ishikawa-Kakiya, Hirotsugu Maruyama.

**Writing – original draft:** Yuki Ishikawa-Kakiya.

**Writing – review & editing:** Yuki Ishikawa-Kakiya, Hirotsugu Maruyama, Kojiro Tanoue, Akira Higashimori, Masaki Ominami, Yuji Nadatani, Shusei Fukunaga, Koji Otani, Shuhei Hosomi, Fumio Tanaka, Yasuhiro Fujiwara.

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
