## [Decision Letter · Decision Letter 0]

6 Sep 2024

PONE-D-24-32776Evaluation of the 22G Franseen needle and 22G Lancet needle for EUS-FNAB sampling in solid pancreatic lesions: Propensity score weightingPLOS ONE

Dear Dr. Maruyama,

Thank you for submitting your manuscript to PLOS ONE. After careful consideration, we feel that it has merit but does not fully meet PLOS ONE’s publication criteria as it currently stands. Therefore, we invite you to submit a revised version of the manuscript that addresses the points raised during the review process.

We look forward to receiving your revised manuscript.

Kind regards,

Kazunori Nagasaka

Academic Editor

PLOS ONE

Journal Requirements:

1. When submitting your revision, we need you to address these additional requirements. Please ensure that your manuscript meets PLOS ONE's style requirements, including those for file naming. The PLOS ONE style templates can be found at https://journals.plos.org/plosone/s/file?id=wjVg/PLOSOne_formatting_sample_main_body.pdf and https://journals.plos.org/plosone/s/file?id=ba62/PLOSOne_formatting_sample_title_authors_affiliations.pdf 2. If any supporting files for review show as item type ‘other’ please change to item type ‘supporting info’ as the reviewer does not have access to these ’other’ files. 3. Please upload a copy of Supporting Information Figure/Table/etc. Supporting information S1 and S2 which you refer to in your text on page 13 and 18.

Additional Editor Comments:

Dear Authors,

Thank you so much for submitting your research to Plos One.

Overall, the manuscript is quite intriguing, and it is worth publishing in Plos One.

However, some concerns have been raised by reviewers.

Please revise the manuscript according to their comments.

We look forward to your revised manuscript soon.

Sincerely,

Kazunori Nagasaka

Reviewers' comments:

Reviewer's Responses to Questions

**Comments to the Author**

1. Is the manuscript technically sound, and do the data support the conclusions?

Reviewer #1: Yes

Reviewer #2: Yes

Reviewer #3: Partly

Reviewer #4: Yes

Reviewer #5: Yes

Reviewer #6: Partly

Reviewer #7: Yes

2. Has the statistical analysis been performed appropriately and rigorously? 

Reviewer #1: Yes

Reviewer #2: Yes

Reviewer #3: Yes

Reviewer #4: No

Reviewer #5: Yes

Reviewer #6: No

Reviewer #7: Yes

3. Have the authors made all data underlying the findings in their manuscript fully available?

Reviewer #1: Yes

Reviewer #2: No

Reviewer #3: Yes

Reviewer #4: Yes

Reviewer #5: Yes

Reviewer #6: No

Reviewer #7: Yes

4. Is the manuscript presented in an intelligible fashion and written in standard English?

Reviewer #1: Yes

Reviewer #2: Yes

Reviewer #3: Yes

Reviewer #4: Yes

Reviewer #5: Yes

Reviewer #6: Yes

Reviewer #7: Yes

5. Review Comments to the Author

Reviewer #1: THe article is interesting although this topic should be addressed in a RCT to obviate to the risk of selection bias.

I am impressed with the low accuracy and sensitivity in this series, particularly with lancet needes. This is not in keeping with the current literature and with the experience of this reviewer and of the scientific community in general.

The authors should perform several subgroup analyses based on some technical factors, for example the use of suction. These aspects should be commented in the discussion (cite the recent papers PMID: 36657607 and PMID: 35915956 )

THe authors should consider the above reported technical features in the covariates used to build the propensity score model

Several different needles were included under the umbrella definition of lancet needle. THe authors should specify some subgroup analysis based on the exact needle design

Any data on the adequacy for molecular analysis? IHC?

Reviewer #2: This article is a retrospective study evaluating the 22G Franseen and 22G Lancet needles for EUS-FNAB sampling in solid pancreatic lesions. The comparison between the 22G Franseen needle and the 22G Lancet needle for endoscopic ultrasound-guided transection (EUS-TA) of pancreatic tumors has been reported in numerous studies, including randomized controlled trials (RCTs) and pathological evaluation has only been performed in a subset of cases. In addition, several points need to be modified.

Major

1. Please add the proportion of cases in which the final diagnosis was made based on surgical specimens.

2. In Table 3, please add the cut-off value for tumor size and number of puncture.

3. The number of EUS-TA experiences of the endoscopists may affect the accuracy of EUS-TA, and the number of EUS-TA experiences shows a significant difference between the two groups in terms of patient background. Therefore, the factor of number of EUS-TA experiences should be added in Table 3.

4. In Table 3, the puncture route was significantly different in univariate analysis. I recommend adding it as a factor in the multivariate analysis.

5. In the amount of core tissue and blood contamination section, you mentioned that the cases for evaluation of tissue specimens were randomly selected. Please describe in detail what method was used to randomly select the cases.

Minor

1. Ref 12 is a protocol article. Please change to PMID: 35151711.

Reviewer #3: This is a retrospective study aimed to compare two EUS needles: the Franssed needle and the standard Lancet needle. Below are my comments to improve the manuscript:

1) Please change EUS-FNAB in EUS-guided tissue acquisition (EUS-TA).

2) In the abstract and in the text, please specify the study setting (academic hospital?)

3) In the abstract, be consistent when you write "diagnostic yield", "sensitivity" or "accuracy"

4) Study aims and endpoints shoud be clearly listed in the Method section. Your "main outcome measure" was "Histological accuracy". What exactly it means? You should refer to standard definitions of outcome measures reported in the AGA white paper (PMID: 29074447). Moreover, the Lancet needle is a cytology needle (FNA), whereas the Franseen needle is a histological needle (FNB). Therefore, comparing the "histological accuracy" may not be the best way. You should compare, as primary outcome" the "diagnostic accuracy" as defined in the AGA white paper.

5) The same as above for the definition of "adequacy".

6) In the discussion, you should discuss if you found differences between the two needle for the diagnosis of autoimmune pancreatitis, because there are reports suggesting the Franseec needle being superior to standard needle in this setting (cite PMID: 32397913 and PMID: 31654634).

7) Also, you should discuss about the possibility of performing Ki-67 evaluation on FNA and FNB samples. Doing so, you should mention a recent metanalysis suggested that FNB could outperform FNA (cite PMID: 35863518).

8) Finally, a recent metanalysis demonstrated a higher rate of tissue integrity and performance of the wet-suction technique compared with the other aspiration techniques despite a larger amount of blood contamination. You shoud mention and comment this study in the discussion (PMID: 36657607).

Reviewer #4: My comments are mainly about statistical methods and handling of numbers.

The use of the inverse probability of treatment weighting (IPTW) method based on the propensity score is a suitable methodology to assess average treatment effect.

There are, however, several flaws in statistical methods and handling of numbers in this manuscript.

The authors perform ed statistical tests using unpaired t-tests for continuous variables and the chi-square tests for categorical variables.

To test continuous variables among two independent groups, unpaired t-tests (parametric tests) or Mann-Whitney tests (non-parametric tests) should be used properly according to the data distribution following normality tests, e.g., Shapiro-Wirk's or Kolmogorov-Smirnov normality tests.

The modern medical statistics professionals recommend that the Fisher's exact test should always be employed to test 2x2 contingency tables even if the sample size is sufficiently large to obtain "exact" test results. If the contingency tables are larger than 2x2, the Fisher-Freeman-Halton test should be used.

They defined minimum p-values as 0.01 (i.e., p-values smaller than 0.01 are written as "p<0.01). The number of significant digits does not comply with the PLOS ONE's regulations clearly written in "Statistical reporting" section ot the PLOS ONE submission guideline. Check the posting rules.

Overall, inappropriate statistical processing is noticeable. The authors should consult with biostatistics experts or one of the co-authors should be a biostatistician.

Reviewer #5: The authors present the results of a retrospective study performed over 15 years comparing the diagnostic capabilities of two different types of EUS-guided aspiration needles. The research is conducted properly and the paper is well written. In my opinion there are some major shortcomings of this work:

1. The study is retrospective; not all data could be reliably identified.

2. The study period is too large; the practice of EUS-FNAB evolved over this period of time, and the use of the two types of needles was not concomitant - the novel type of needle entered in clinical use only in the last 5 years of the study.

3. The heterogeneity in the study groups is quite large; for example more than one type of Lancet needle was used, which could affect the overall results.

4. The study center has a low volume of procedures. This results from the fact that 315 patients were enrolld over 15 years, which means that a mean of less than 21 patients were enrolled per year in both arms.

5. The study is single center, in a center where the diagnostic accuracy using the novel types of FNB needles is 10% less than expected as recommeded by quality in endoscopy guidelines, therefore the center's endoscopic/histopathologic practice needs to be improved.

6. On the sudy overall set of data, a statistical method needed to be used (the inverse probability of treatment weighting method), and it is not clear why. Probably because the study groups are not equivalent.

7. Postive EUS-FAB results alone could classify a patient as positive, which could negatively impact upon the calculation of specificity and accuracy.

8. For the evaluation of core tissue and blood contamination it was not possible to collect slides for all cases, therefore only30 cases were randomly selected and the slides were evaluated.

9. The study groups are not equivalent: 50 patients (20%) were excluded from the first group and only 1 (1%) from the second group; tumor size was significantly larger in the Franseen needle group than in Lancet needle group (p<0.01); the number of expert endoscopists was significantly higher in first group (p<0.01). The same for location of the tumor, needle puncture route and final diagnoses.

10. The study results are not new. Almost all previous studies, some prospective, concluded that Franseen needle obtained higher amount of tissue, which led to its high diagnostic accuracy.

11. There was no dedicated protocol for cytology using the Fanseen needles, such as touch imprint cytology; the sample is generally processed as a histological sample (i.e. fixed in formalin), and not as a cytological sample.

Reviewer #6: (General comment) The manuscript should closely follow RECORD Guideline https://www.record-statement.org/.

(Title) Please avoid the use of abbreviations in the title of the manuscript.

(Abstract) "were evaluated using a scoring system" which scoring system? Please be specific.

(Abstract) "were divided into the Lancet (n=200) and Franseen needle group (n=115)" Which criterion was used?

(General comment) Report the p-values with at least three decimals.

(Abstract) "blood contamination scores of 3" it is unclear what a score of 3 means.

(Introduction) "was significantly higher than that of Lancet needle" please be specific and write the ranges of the reported values with the associated 95% confidence interval.

(Introduction) "accuracy is limited" please be specific.

(Introduction) "amounts of tissue core were evaluated in some reports" were comparable? Please be specific.

(Introduction) "would be superior to that of a Lancet needle" explain why.

(Introduction) The state of the art is not presented in sufficient details to capture the gaps in the scientific literature.

(Methods) During 2006 and 2021 changes in needles manufacturing could be seen. Furthermore, different manufacturers had different quality for the same medical device.

(Methods) "insufficient data" which data? Please be specific.

(Methods) The performance also depend by the experience of the physician. How did you treat this aspect in your study.

(Methods) "All the patients provided written informed consent for the use of their personal data." "This retrospective comparative study was conducted at a single center." It is unclear how in a study with retrospective data collection the patients gave their consent.

(Methods) It is unclear which was the gold standard diagnosis.

(Methods) Define ROSE abbreviation.

(Methods) "A thin specimen was placed on a slide and air-dried" When? By whom?

(Methods) "30 cases were randomly selected and the slides were checked" this is insufficient. Why 30? The variability in a such a small sample is not representative for the eligible population.

(Methods) Define "few", "moderate" and "high".

(Methods) "If the sample size was too small, Fisher’s exact test" this is out of statistical knowledge.

(Methods) Without a gold standard diagnostic your study is just a statistical exercise.

(Methods) "Kappa coefficients" is not appropriate according with the described methodology.

(Methods) "receiving the exposure that the patients" define "exposure".

(Methods) Generally the SPSS has a one-year license and 21 is far from the current version.

(Methods) It is unclear how age and sex "might influence the diagnostic".

(Methods) The details presented in this section does not allow the reproduction/replication.

(Methods) It is unclear who chose the type of needle.

(Methods) Please provide the date for ethical approval.

(Results) "followed up" no information regarding the follow up was provided in the Methods section.

(Results) IQR with round brackets means that the values are not included in the range. Is this correct?

(Results) The point estimators in table 2 must have the 95% CI

(Results) How the variables were selected for multivariate analysis. Why route is not in the model.

(Results) Also report the 95% CI for κ-value.

(Results) List adverse effect for each needle with its distribution.

(Results) The quality of the figure are low.

(Discussion) I was not able to see any reference to your own results (tables).

(Discussion) Put the limitations in the order of their importance.

Reviewer #7: In this original article entitled “Evaluation of the 22G Franseen needle and 22G Lancet needle for EUS-FNAB sampling in solid pancreatic lesions: Propensity score weighting,” Yuki Ishikawa-Kakiya et al. conducted a retrospective comparative study aimed to evaluate the histologic accuracy of two needles in pancreatic lesions.

The study is well written, with a clear and concise presentation and detailed method section. The use of a single technique and the relatively low number of patients included are limitations of the study. However, using a propensity score has mitigated the bias, and the results align with the literature. I have some suggestions:

-In the section “true and false diagnosis” the inadequate sample definition is confounding (line 143). Not adequate and not accurate are different definitions, as correctly signed in the following article paragraphs.

-in the discussion the impact of the size of the lesion and the benign nature (especially Autoimmune pancreatitis) should be cited.

-please correct the typo Franssen (line 308).

The article may be accepted after minor revisions.

6. PLOS authors have the option to publish the peer review history of their article (what does this mean? ). If published, this will include your full peer review and any attached files.

**Do you want your identity to be public for this peer review?** For information about this choice, including consent withdrawal, please see our Privacy Policy .

Reviewer #1: No

Reviewer #2: No

Reviewer #3: No

Reviewer #4: No

Reviewer #5: No

Reviewer #6: No

Reviewer #7: No

---

## [Author Response · Author response to Decision Letter 1]

30 Dec 2024

Thank you very much for reviewing our manuscript. As the responses cover multiple points, please refer to the attached file for details.

Point-by-Point Responses (PONE-D-32776)

REVIEWER'S COMMENTS TO THE AUTHOR:

Reviewer #1

Comment 1:

The article is interesting although this topic should be addressed in a RCT to obviate to the risk of selection bias.

Responses to the Reviewer’s comments:

Thank you for your insightful comments. We completely agree that a randomized controlled trial (RCT) would be the best approach, and some studies have already reported before. In those studies, the accuracy of Franseen needle was higher than traditional FNA needle [1-3]. However, tissue obtained from EUS-FNAB is traditionally evaluated using cytology and histology, which has not been discussed about the details of diagnosis method; cytological, histological or combined accuracy is limited in the Franseen needle and Lancet needle. In this study, we compared various diagnostic methods, with a particular focus on histological diagnosis. In addition, we examined the amount of blood contamination, effect of which has not been sufficiently evaluated in previous studies.

This comparison was conducted through a retrospective observational study in order to increase the sample size. We acknowledge the potential issue of selection bias inherent in retrospective studies. To address this, we used propensity score weighting with the IPTW method to minimize bias and make the analysis more akin to a pseudo-RCT. We hope this approach is satisfactory and meets your expectations.

Comment 2:

I am impressed with the low accuracy and sensitivity in this series, particularly with lancet needles. This is not in keeping with the current literature and with the experience of this reviewer and of the scientific community in general.

Responses to the Reviewer’s comments:

Thank you for your insightful comments. We acknowledge that the diagnostic accuracy and sensitivity of the lancet needle in our study are lower than those reported in recent literature and may not align with the broader experience of the scientific community.

However, some previous reports have also shown a lower diagnostic accuracy for the lancet needle, with rates as low as 63.5%[4]. In our study, this lower accuracy may be partially attributed to the inclusion of earlier cases of EUS-FNA performed at our institution, when both endoscopic technique and cytological evaluation were still in their developmental stages. This learning curve effect likely impacted the overall diagnostic performance of the lancet needle.

Another important factor may be the amount of sample obtained. The amount of tissue collected with the lancet needle was often small. Small amounts of tissue increase the possibility of histological misinterpretation, especially in the presence of inflammation, which may mask or mimic pathological findings and reduce diagnostic accuracy. To examine this issue, we performed a sub-analysis limited to cases from 2016 onwards (n = 196), when the Franceen needle became available. The diagnostic accuracy of the Lancet needle was 64.3%, and that of the Franseen needle was 85.8%. The diagnostic accuracy of both the Lancet needle and Franseen needle has increased, which may have improved the skills of the technologists and endoscopists. In addition, univariate and multivariate analyses were performed on cases from 2016 onwards, and the tissue diagnostic rate of the FNB needle was shown to be significantly better.

Univariate Multivariate

　 Crude OR (95% CI|) p value 　 Crude OR (95% CI) p value

Needle, Franseen 2.88 (1.43–5.82) 0.003 2.42 (1.17-5.02) 0.018

Number of punctures 0.63 (0.45–0.89) 0.008 　 0.71 (0.50-1.17)　 0.056

Comment 3:

The authors should perform several subgroup analyses based on some technical factors, for example the use of suction. These aspects should be commented in the discussion (cite the recent papers PMID: 36657607 and PMID: 35915956 )

The authors should consider the above reported technical features in the covariates used to build the propensity score model

Responses to the Reviewer’s comments:

Thank you for your valuable suggestions. I agree that technical factors such as aspiration technique may affect diagnostic results. Recent studies have demonstrated that wet-suction, both with FNB and FNA needles, can provide better tissue integrity, higher cell density, and better diagnostic accuracy compared with standard aspiration and slow pull techniques[5] [6]. These findings highlight the potential impact of the suction method on EUS-FNAB performance. However, as our study is retrospective in nature, detailed information regarding the aspiration method (e.g., wet suction or suction) was not consistently recorded across all cases. This lack of data precluded us from conducting subgroup analyses based on these technical factors. We acknowledge this as a limitation of our study and have clearly addressed it in the revised manuscript (page 30, line 403-409), “First, we could not evaluate the relationship between factors such as the utilization of suction method (general suction or wet-suction), however, these factors may potentially influence diagnostic accuracy. Previous reports have indicated that wet suction resulted in blood contamination similar to that of the general suction method, but yielded better results in terms of diagnostic rate and tissue quality. Similarly, the puncture technique such as door knocking method or fanning method may affect diagnostic accuracy; however, due to insufficient data on this factor, it could not be fully evaluated in this study.”. We greatly appreciate the reviewer’s insightful comments, which will guide future studies. Moving forward, prospective studies with standardized documentation of technical variables, including suction methods, are warranted to better elucidate their impact on diagnostic outcomes.

Comment 4:

Several different needles were included under the umbrella definition of lancet needle. The authors should specify some subgroup analysis based on the exact needle design

Responses to the Reviewer’s comments:

Thank you very much for your valuable comments. As you mentioned, the sono tip and expect are lancet needles, while the EZshot3plus is a Menghini needle [7]. However, all of these are considered FNA needles, and it has been reported that the EZ shot3 plus is included in the lancet needle group [8]. The Menghini needle is said to have a sharper lancet tip, which improves puncture performance. Therefore, I believe there is no issue with the classification. Furthermore, if we were to conduct subgroup analysis comparing lancet needles and Menghini needles, the sample size would be very small, making the results less reliable. However, we will consider adding further clarification or subgroup analysis if necessary in the revised manuscript.

Comment 5:

Any data on the adequacy for molecular analysis? IHC?

Responses to the Reviewer’s comments:

Thank you for your comments. Unfortunately, in our institution, molecular analysis and immunohistochemistry are not routinely performed in the majority of cases. As such, we are unable to provide data or discuss for these analyses in the current study. However, we recognize that the critical role of IHC and molecular markers, such as Ki-67, in enhancing diagnostic accuracy and guiding treatment decisions.

There are reports that Ki-67 evaluation in EUS-FNA samples is a useful tool for excluding G3 tumors, and that immunohistochemical staining for p53 and Ki-67 can improve the sensitivity of EUS-FNA in diagnosing pancreatic adenocarcinoma[9, 10]. Immunostaining in EUS-FNA is therefore extremely important, and a comparative study between FNA and FNB in this context would be highly valuable. A recent meta-analysis also suggested that EUS-FNB might outperform EUS-FNA in terms of obtaining adequate tissue for accurate Ki-67 assessment[11]. This finding indicates that EUS-FNB could provide a more reliable method for tumor grading, thereby enhancing diagnostic precision and potentially reducing the risk of over- or undertreatment. While we were unable to assess these aspects in our current study due to institutional limitations, we agree that further investigations comparing EUS-FNA and EUS-FNB for molecular and IHC adequacy would be highly valuable. We appreciate this suggestion, which highlights an important area for future research.

Reviewer #2

Comment 1 (major):

Please add the proportion of cases in which the final diagnosis was made based on surgical specimens.

Responses to the Reviewer’s comments:

Thank you very much for your valuable comments. The percentage of patients who underwent surgery was 43% in the Lancet needle group and 29.6% in the Franseen needle group, with a statistically significant difference between in the two group (p = 0.02). We added the data to table1, as follow.

We appreciate your insightful suggestion, which has improved the completeness of our study.

Table 1. Clinical characteristics of the Lancet and Franseen needle groups.

　 Lancet needle (n=200) Franseen needle (n=115) p value

Median age (IQR†), years 69 [60–74] 71 [66–76] 0.009

Sex, male, n (%) 115 (57.5) 62 (53.9) 0.557

Median tumor size, mm (IQR†) 22 [16.1–27.9] 27 [20.9–32.8] <0.001

Location of the lesion, n (%) 0.008

Head 119 (59.5) 52 (45.2)

Body 57 (28.5) 34 (29.6)

Tail 22 (11.0) 28 (24.3)

All 2 (1.0) 1 (0.9)

Puncture route, n (%) 0.001

Stomach 85 (42.5) 68 (59.1)

Duodenal bulb 57 (28.5) 31 (27.0)

Duodenal second portion 58 (29) 15 (13.0)

Jejumum 0 1 (1.0)

Scopist, expert (%) 65 (32.5) 11 (9.6) <0.001

Median number of passes (IQR†) 2 [2–3] 2 [1–2] 0.003

Adverse events, n (%) 3 (1.5) 3 (2.6) 0.672

Surgery 86 (43.0) 34 (29.5) 0.021

Final diagnosis, n (%)

Malignant 139 (69.5) 91 (79.1) 0.067

Pancreatic carcinoma 126 82

Neuroendocrine carcinoma 1 1

Metastatic cancer 5 4

Intraductal papillary mucinous with high-grade dysplasia 1 0

Intraductal papillary mucinous with invasive carcinoma 1 2

Malignant lymphoma 4 1

Carcinoma of unknown primary 1 1

Benign 61 (30.5) 24 (20.9)

Autoimmune pancreatitis 12 11

Neuroendocrine neoplasm 15 5

Tumor-forming pancreatitis 2 0

Serous cystic neoplasm 5 1

Non-specific inflammation 14 2

Chronic pancreatitis 6 2

Intraductal papillary mucinous neoplasm 2 2

Pancreatic pseudocyst 2 0

Lymphoepitelial cyst 0 1

Intrapancreatic accessory spleen 3 0 　

Adequate sample, % 182 (91.0) 111 (96.5) 0.069

†: IQR, interquartile range.

Comment 2:

In Table 3, please add the cut-off value for tumor size and number of puncture.

Responses to the Reviewer’s comments:

Thank you for your insightful comments. As you pointing out, we calculated the cutoff value. Regarding to number of punctures, an AUC of 0.6 indicates that the model's diagnostic performance was modest, with a 60% probability of correctly distinguishing between cases. This suggests that while two punctures may enhance diagnostic yield to some extent, factors other than the number of punctures, such as needle type, target lesion characteristics, or operator experience, may have a significant influence on diagnostic accuracy. In the other hand, the AUC for tumor size was 0.5 with a cutoff value of 30 mm, indicating that tumor size alone is not a decisive diagnostic factor. We have added this information to Table 3, as follow.

Table 3. Prognostic factor for histological accuracy.

Univariate Multivariate

　 Crude OR¶¶ (95% CI||||) p value 　 Crude OR¶¶ (95% CI||||) p value

Age 1.00 (0.98–1.03) 0.656

Sex, male 1.03 (0.59–1.81) 0.911 　 　 　

Tumor size 1.00 (0.97–1.02) 0.767 　 　 　

Puncture route, duodenum 1.07 (0.53–0.77) 0.122 　 1.17 (0.83-1.65)　 0.379　

Number of punctures 1.47 (1.14–1.88) 0.003 　 1.35 (1.04–1.75) 0.022

Needle, Franseen 2.29 (1.21–4.35) 0.011 　 1.85 (0.95–3.61) 0.069

AUC, number of puncture (95% CI) 61.9 (0.54 - 0.69)

AUC, tumor size (95% CI) 51.0 (0.44 - 0.58)

¶¶: OR, odd ratio, ||||: CI, confidence interval

Comment 3:

The number of EUS-TA experiences of the endoscopists may affect the accuracy of EUS-TA, and the number of EUS-TA experiences shows a significant difference between the two groups in terms of patient background. Therefore, the factor of number of EUS-TA experiences should be added in Table 3.

Responses to the Reviewer’s comments:

Thank you for your insightful comments. Previous reports have indicated that there is no difference in diagnostic accuracy based on the skill level of the operator, nor is there a difference in the collection of tissue suitable for comprehensive genomic profiling according to the operator’s level[12]. Furthermore, Mukai et al. reported that there were no technical failures even when practitioners used the Franseen needle[13]. However, the number of endoscopic procedures performed could indeed affect diagnostic accuracy as you pointing out.

In our institution, an expert always supervises the procedure. If a less experienced endoscopist performs the EUS-TA and is unable to obtain a sufficient tissue sample, the expert takes over and performs the puncture. Therefore, we believe this minimizes the impact of operator experience on the results, and for this reason, we did not include it in the logistic regression analysis.

That said, if you believe it is necessary to include this factor in the analysis, we would be happy to consider performing additional analyses.

Comment 4:

In Table 3, the puncture route was significantly different in univariate analysis. I recommend adding it as a factor in the multivariate analysis.

Responses to the Reviewer’s comments:

Thank you for your valuable comments. As you pointed out, the puncture route may potentially affect the diagnostic accuracy[14]. Previous studies have shown improved histological diagnostic rates with the transgastric route. There was no impact of puncture route differences on the diagnostic accuracy of cytology; however, histological diagnosis showed higher accuracy with the transgastric route compared to the transduodenal route (85.2% vs 76.5) [15]. This is likely due to the greater difficulty of obtaining samples from the transduodenal route, resulting in a smaller sample volume and lower diagnostic accuracy. Taking these reports into account,

we performed a multivariate analysis including the puncture route, and the results are shown in Table 3. We also included both the puncture route and the number of punctures in the multivariate analysis, and it became clear that the Franseen needle remained a significant factor for obtaining a histological diagnosis.

In consideration of your comments, I have added the following to the Discussion section, as follow: Previous reports have indicated that the number of punctures and puncture route are factors that influence diagnostic accuracy. In the present study, we adjusted for these factors using multivariate analysis, and we found that the diagnostic accuracy of the Franseen needle was still higher compared to the lancet needle (page 26, line 330-333).

Table 3. Prognostic factor for histological accuracy.

Univariate Multivariate

　 Crude OR¶¶ (95% CI||||) p value 　 Crude OR¶¶ (95% CI||||) p value

Age 1.00 (0.98–1.03) 0.656

Sex, male 1.03 (0.59–1.81) 0.911 　 　 　

Tumor size 1.00 (0.97–1.02) 0.767 　 　 　

Puncture route, duodenum 1.07 (0.53–0.77) 0.122 　 1.17 (0.83-1.65)　 0.379　

Number of punctures 1.47 (1.14–1.88) 0.003 　 1.35 (1.04–1.75) 0.022

Needle, Franseen 2.29 (1.21–4.35) 0.011 　 1.85 (0.95–3.61) 0.069

AUC, number of puncture (95% CI) 61.9 (0.54 - 0.69)

AUC, tumor size (95% CI) 51.0 (0.44 - 0.58)

¶¶: OR, odd ratio, ||||: CI, confidence interval

Comment 5:

In the amount of core tissue and blood contamination section, you mentioned that the cases for evaluation of tissue specimens were randomly selected. Please describe in detail what method was used to randomly select the cases.

Responses to the Reviewer’s comments:

Thank you for your insightful comments. This study was conducted as a retrospective design. Due to the inability to collect slides from older cases, we randomly ext

---

## [Decision Letter · Decision Letter 1]

30 Jan 2025

PONE-D-24-32776R1Evaluation of the ��G Franseen needle and ��G Lancet needle for endoscopic ultrasonography-guided tissue acquisition sampling in solid pancreatic lesions: Propensity score weightingPLOS ONE

Dear Dr. Maruyama,

Thank you for submitting your manuscript to PLOS ONE. After careful consideration, we feel that it has merit but does not fully meet PLOS ONE’s publication criteria as it currently stands. Therefore, we invite you to submit a revised version of the manuscript that addresses the points raised during the review process.

We look forward to receiving your revised manuscript.

Kind regards,

Kazunori Nagasaka

Academic Editor

PLOS ONE

Journal Requirements:

Additional Editor Comments:

Dear Authors,

Please revose the manuscript accoring to the comments (Statistical process, and some typos).

I look forward to receiveing your revised manuscript soon.

Sincerely,

Kazunori Nagasaka

Reviewers' comments:

Reviewer's Responses to Questions

**Comments to the Author**

1. If the authors have adequately addressed your comments raised in a previous round of review and you feel that this manuscript is now acceptable for publication, you may indicate that here to bypass the “Comments to the Author” section, enter your conflict of interest statement in the “Confidential to Editor” section, and submit your "Accept" recommendation.

Reviewer #1: All comments have been addressed

Reviewer #4: All comments have been addressed

2. Is the manuscript technically sound, and do the data support the conclusions?

Reviewer #1: Yes

Reviewer #4: Yes

3. Has the statistical analysis been performed appropriately and rigorously? 

Reviewer #1: Yes

Reviewer #4: Yes

4. Have the authors made all data underlying the findings in their manuscript fully available?

Reviewer #1: (No Response)

Reviewer #4: Yes

5. Is the manuscript presented in an intelligible fashion and written in standard English?

Reviewer #1: Yes

Reviewer #4: Yes

6. Review Comments to the Author

Reviewer #1: The authors improved the manuscript according to my suggestions. Now the manuscript is ready for publication.

I don't have other suggestions. Thank you!

Reviewer #4: I pointed out some flaws with statistical processing and reporting in my previous review. The authors have examined and corrected them as appropriate.

One minor correction: in P11L193, "should be used." should be "was used."

7. PLOS authors have the option to publish the peer review history of their article (what does this mean? ). If published, this will include your full peer review and any attached files.

**Do you want your identity to be public for this peer review?** For information about this choice, including consent withdrawal, please see our Privacy Policy .

Reviewer #1: No

Reviewer #4: **Yes: ** Ryuichiro ARAKI

---

## [Author Response · Author response to Decision Letter 2]

8 Feb 2025

Point-by-Point Responses (PONE-D-24-32776R2)

Dear Editor and Reviewers,

Thank you for your valuable feedback and for reviewing our manuscript. We sincerely appreciate your constructive comments, which have helped us improve the quality of our paper. We have carefully addressed all the comments and revised the manuscript accordingly.

Below, we provide our detailed responses to each comment and highlight the corresponding changes in the manuscript.

Reviewer #4's comment:

"One minor correction: in P11L193, 'should be used.' should be 'was used.'"

Response:

Thank you for your suggestion. We have corrected this phrase accordingly (P11L193).

Regarding the reference list, we have carefully reviewed all cited papers and confirmed that no retracted papers are included.

We have carefully reviewed the statistical processing section, and as Reviewer #4 has acknowledged, the necessary corrections have already been made appropriately in the previous revision. Given that both Reviewer #1 and Reviewer #4 have confirmed that our statistical analysis is now appropriate and rigorous, we believe that no further modifications are required in this regard.

If the Editor still requires additional revisions related to statistical processing, we would appreciate specific guidance on the necessary changes.

We hope that the revisions adequately address your concerns.

Thank you again for your time and effort in reviewing our manuscript.

---

## [Editor Report · Decision Letter 2]

16 Feb 2025

PONE-D-24-32776R2Evaluation of the 22G Franseen needle and 22G Lancet needle for endoscopic ultrasonography-guided tissue acquisition sampling in solid pancreatic lesions: Propensity score weightingPLOS ONE

Dear Dr. Maruyama,

Thank you for submitting your manuscript to PLOS ONE. After careful consideration, we feel that it has merit but does not fully meet PLOS ONE’s publication criteria as it currently stands. Therefore, we invite you to submit a revised version of the manuscript that addresses the points raised during the review process.

We look forward to receiving your revised manuscript.

Kind regards,

Kazunori Nagasaka

Academic Editor

PLOS ONE

Journal Requirements:

Additional Editor Comments:

Please modify the text according to Reviewer 2's comments. After the necessary modifications are made, we hope to proceed towards a decision of acceptance.

---

## [Author Response · Author response to Decision Letter 3]

24 Mar 2025

Mar 24, 2025

Emily Chenette

Editors-in-Chief

Associate Editors

PLOS ONE

Dear Editor

We appreciate the opportunity to submit a revised version of our manuscript entitled, “Evaluation of the 22G Franseen needle and 22G Lancet needle for endoscopic ultrasonography-guided tissue acquisition sampling in solid pancreatic lesions: Propensity score weighting.” (manuscript ID: PONE-D-24-32776R3) for consideration for publication in PLOS ONE.

We believe that we have addressed all concerns raised by the reviewers as detailed in the accompanying point-by-point responses.

All authors concur with the submission of this manuscript. We ascertain that none of the data in this manuscript have been previously reported, nor is the manuscript under consideration for publication elsewhere.

We hope that PLOS ONE now finds our manuscript suitable for publication. We appreciate your consideration of our work.

Sincerely Yours,

Hirotsugu Maruyama

Hirotsugu Maruyama, M.D. PhD.

Department of Gastroenterology

Osaka City University Graduate School of Medicine

1-4-3, Asahimachi, Abeno-ku, Osaka-City, Osaka, 545-8585, Japan

e-mail to; hiromaruyama99@gmail.com

Phone: +81-6-6645-3811

FAX: +81-6-6645-3813

Point-by-Point Responses (PONE-D-24-32776R3)

Reviewer #2's comment:

2. In Table 3, please add the cut-off value for tumor size and number of punctures.

Response:

I apologize for not responding adequately to your previous comments. I had not provided sufficient details regarding the cut-off value, therefore I would like to address them again as follows.

As you pointing out, we calculated the cut-off value. Regarding number of punctures, the cut-off value was three and an AUC of 0.6 indicates that the model's diagnostic performance was modest, with a 60% probability of correctly distinguishing between cases. This suggests that while three punctures may enhance diagnostic yield to some extent, factors other than the number of punctures, such as needle type, target lesion characteristics, or operator experience, may have a significant influence on diagnostic accuracy. In the other hand, the AUC for tumor size was 0.5 with a cut-off value of 29 mm, indicating that tumor size alone is not a decisive diagnostic factor. Additionally, I have included information about the cut-off value in Table 3. Please review the revisions.

3. The number of EUS-TA experiences of the endoscopists may affect the accuracy of EUS-TA, and the number of EUS-TA experiences shows a significant difference between the two groups in terms of patient background. Therefore, the factor of number of EUS-TA experiences should be added in Table 3.

Response:

I apologize for not responding adequately to your previous comments, therefore I would like to address them again as follows.

Thank you for your insightful comments. Previous reports have indicated that there is no difference in diagnostic accuracy based on the skill level of the operator, nor is there a difference in the collection of tissue suitable for comprehensive genomic profiling according to the operator’s level. Furthermore, Mukai et al. reported that there were no technical failures even when practitioners used the Franseen needle. However, the number of endoscopic procedures performed could indeed affect diagnostic accuracy as you pointing out.

We performed both univariable (OR, 1.204; 95% CI, 0.64-2.29; p=0.566) and multivariable analyses (OR, 0.89; 95% CI, 0.45-1.75; p=0.733) to assess whether expertise influenced accuracy, however, no significant results indicating an impact on diagnosis were obtained. In our institution, an expert always supervises the procedure. If a less experienced endoscopist performs the EUS-TA and is unable to obtain a sufficient tissue sample, the expert takes over and performs the puncture. Therefore, we believe this minimizes the impact of operator experience on the results. Additionally, we were concerned that increasing the number of variables could reduce the precision of the multivariable analysis. Therefore, in Table 3 of the manuscript, we have included only the results of the univariable analysis. Then, we added this sentence into the discussion section ‘Subsequently, we evaluated the impact of expert experience on diagnostic accuracy. Previous reports suggest that the skill level of the endoscopist does not significantly affect diagnostic accuracy[27]. Similarly, our findings were consistent with this. One possible reason for this is that all procedures at our institution are performed under the supervision of an expert.’ Please review the manuscript, line of 340-343.

27. Ishigaki K, Nakai Y, Endo G, Kurihara K, Ishida K, Tange S, et al. Feasibility of comprehensive genomic profiling using endoscopic ultrasound-guided tissue acquisition with a 22-gauge Franseen needle. DEN Open. 2024;4(1):e365.

---

## [Decision Letter · Decision Letter 3]

30 Mar 2025

Evaluation of the 22G Franseen needle and 22G Lancet needle for endoscopic ultrasonography-guided tissue acquisition sampling in solid pancreatic lesions: Propensity score weighting

PONE-D-24-32776R3

Dear Dr. Maruyama,

We’re pleased to inform you that your manuscript has been judged scientifically suitable for publication and will be formally accepted for publication once it meets all outstanding technical requirements.

Kind regards,

Kazunori Nagasaka

Academic Editor

PLOS ONE

Additional Editor Comments (optional):

Dear Authors,

We are pleased to inform you that your revised manuscript has satisfactorily addressed all reviewer comments.

Your responses were thorough, clear, and effectively incorporated into the manuscript. Consequently, we find the paper now meets the publication standards of PLOS ONE.

Therefore, your manuscript is accepted for publication. Congratulations on your successful revisions and valuable contribution.

Thank you for choosing PLOS ONE for your work.

We look forward to seeing your paper published.

Sincerely,

Kazunori Nagasaka

Reviewers' comments:

Reviewer's Responses to Questions

**Comments to the Author**

1. If the authors have adequately addressed your comments raised in a previous round of review and you feel that this manuscript is now acceptable for publication, you may indicate that here to bypass the “Comments to the Author” section, enter your conflict of interest statement in the “Confidential to Editor” section, and submit your "Accept" recommendation.

Reviewer #2: All comments have been addressed

2. Is the manuscript technically sound, and do the data support the conclusions?

Reviewer #2: Yes

3. Has the statistical analysis been performed appropriately and rigorously? 

Reviewer #2: Yes

4. Have the authors made all data underlying the findings in their manuscript fully available?

Reviewer #2: Yes

5. Is the manuscript presented in an intelligible fashion and written in standard English?

Reviewer #2: Yes

6. Review Comments to the Author

Reviewer #2: The revised manuscript has appropriately addressed all comments raised by the reviewer. The authors provided thorough and clear responses, effectively incorporating necessary revisions into the manuscript. Therefore, I am satisfied with the changes made and have no further comments or additional concerns to raise at this stage.

7. PLOS authors have the option to publish the peer review history of their article (what does this mean? ). If published, this will include your full peer review and any attached files.

**Do you want your identity to be public for this peer review?** For information about this choice, including consent withdrawal, please see our Privacy Policy .

Reviewer #2: No

---

## [Editor Report · Acceptance letter]

PONE-D-24-32776R3

PLOS ONE

Dear Dr. Maruyama,

I'm pleased to inform you that your manuscript has been deemed suitable for publication in PLOS ONE. Congratulations! Your manuscript is now being handed over to our production team.

Kind regards,

on behalf of

Professor Kazunori Nagasaka

Academic Editor

PLOS ONE